# MVDR-LSTM Distance Estimation Model Based on Diagonal Double Rectangular Array

**DOI:** 10.3390/s23115094

**Published:** 2023-05-26

**Authors:** Xiong Zhang, Wenbo Wu, Jialu Li, Fan Dong, Shuting Wan

**Affiliations:** 1Hebei Key Laboratory of Electric Machinery Health Maintenance & Failure Prevention, North China Electric Power University, Baoding 071003, China; hdjxzx@ncepu.edu.cn; 2Hebei Engineering Research Center for Advanced Manufacturing & Intelligent Operation and Maintenance of Electric Power Machinery, North China Electric Power University, Baoding 071003, China; 3Department of Mechanical Engineering, North China Electric Power University, Baoding 071003, China; w15031435169@163.com (W.W.); ljl2481390881@163.com (J.L.); df20000109@163.com (F.D.)

**Keywords:** diagonal double rectangular array, MVDR, LSTM, idler failure distance estimation

## Abstract

Deep learning algorithms have the advantages of a powerful time series prediction ability and the real-time processing of massive samples of big data. Herein, a new roller fault distance estimation method is proposed to address the problems of the simple structure and long conveying distance of belt conveyors. In this method, a diagonal double rectangular microphone array is used as the acquisition device, minimum variance distortionless response (MVDR) and long short-term memory network (LSTM) are used as the processing models, and the roller fault distance data are classified to complete the estimation of the idler fault distance. The experimental results showed that this method could achieve high-accuracy fault distance identification in a noisy environment and had better accuracy than the conventional beamforming algorithm (CBF)-LSTM and functional beamforming algorithm (FBF)-LSTM. In addition, this method could also be applied to other industrial testing fields and has a wide range of application prospects.

## 1. Introduction

In recent years, due to the increasing emphasis on energy conservation and emission reduction, straw power plants have developed rapidly. As an important piece of equipment for conveying materials, the safety of idlers in belt conveyors has a crucial impact on the smooth operation of the entire conveying system. As the key support and rotating component of the belt conveyor, the chain reaction caused by the failure of the idler poses a safety threat to the entire straw power station. The long-term operation, substantial conveying distances, and huge weight of idlers in harsh working environments necessitate daily inspections and regular maintenance that not only entail significant time and money costs, but also allow for some seriously degraded parts to be overdue for replacement. Therefore, it is of great significance to design non-contact acoustic signal measurement and fault distance estimation methods suitable for belt conveyor idlers in power stations, so as to reasonably arrange maintenance activities; reduce time, manpower, and material costs; and ensure the reliability and safety of mine belt conveyor operation.

In the field of idler bearing fault diagnosis, a method of energy collection was proposed in [1] that was of certain practical value for bearing fault diagnosis. The authors of [2] proposed an electromagnetic energy collector for bearing fault diagnosis that improved the diagnosis efficiency and reduced the difficulty of diagnosis.

With the acceleration of industrial processes, the safety of machinery and equipment has become the top priority, and methods for the fault diagnosis and condition detection of machinery and equipment are also constantly improving and upgrading. In recent years, the use of deep learning to diagnose bearings has also become a popular research direction. Zhou proposed a novel combination of discrete probabilistic entropy-based health indicators (HIs) and long short-term memory (LSTM)-based methods to predict bearing health. The LSTM model included in the proposed method could accurately predict bearing health [3]. Aljemely et al. designed a combination of long short-term memory and large margin nearest neighbor (LSTM-LMNN), which improved work efficiency and diagnostic accuracy and was superior to traditional diagnostic methods in terms of stability and reliability [4]. In order to increase the effectiveness of convolutional neural networks (CNNs) for bearing fault feature recognition, Xie et al. proposed four different CNN hybrid models, and after an experimental comparison, the CNN-random forest (RF) and CNN-support vector machine (SVM) algorithms were found to make full use of CNN feature extraction capabilities [5]. Yang et al. proposed a method based on the combination of variable mode decomposition (VMD) and a CNN. Firstly, after collecting the motor bearing fault signal, the motor bearing data were denoised through VMD, and the denoising data were extracted through a CNN for diagnosis. Experiments showed that the accuracy of VMD-CNN for motor bearing fault diagnosis was better than that of a CNN [6]. Yu et al. proposed a novel real-time monitoring method for sediment particles, combining a stepoelectric nanogenerator driven by particle droplets (PLDD-TENG) and deep learning technology. By analyzing the output signal of PLDD-TENG, this method could sensitively reflect different particle types and concentrations [7].

Beamforming suppresses signals in non-selected directions and enhances signals in selected directions by merging the signals collected by the microphone array; in addition, it can realize focused pickup in the specified direction, which can effectively improve the signal-to-noise ratio of the received signal. As beamforming algorithms have become more popular, they have also played an increasingly important role in the field of sound source identification [8,9,10,11]. MVDR, a commonly used beamforming algorithm, is widely used in the field of fault diagnosis. The main advantages of the MVDR algorithm are summarized below.

Efficient fault feature extraction: The MVDR algorithm can effectively extract fault signal features. Due to its distortion-free nature, the MVDR algorithm can minimize noise and interference while retaining the fault signal, thereby improving the accuracy of fault detection.

Excellent spatial resolution: In terms of fault source location, the high spatial resolution of the MVDR algorithm can provide more accurate fault source location information.

Adaptability: In complex industrial environments, troubleshooting conditions can change. As an adaptive algorithm, MVDR can automatically adjust parameters according to changes in the environment, so as to effectively diagnose faults under various conditions.

Robustness: The MVDR algorithm has good robustness for independent and uniformly distributed noise signals, which allows it to maintain good fault diagnosis performance in various noise environments.

Subramanian et al. used a deep learning neural network to identify and locate multiple sound sources on the basis of direction of arrival estimation, and experimental results showed that the proposed method could classify DOAs with high resolution [12]. Considering the problem of far-field speech processing, Zhang et al. proposed a deep special beamforming technology that combined a microphone array with multi-channel speech enhancement based on deep learning and significantly reduced the probability of a far-field acoustic environment [13]. Yang et al. proposed an MVDR algorithm based on deep learning for extracting target speakers from hybrid speech that first used the target voice signal for estimation as the basis for channel selection and then applied the selected signal to the MVDR algorithm based on deep learning to extract the target speaker from hybrid speech. This method has achieved good results in both simulation and practical applications [14]. Ramezanpour and Mosavi proposed a neural beamformer for estimating the desired signal in noisy and interference environments that consisted of a convolutional neural network (CNN) and bidirectional long short-term memory (BiLSTM) to estimate the interference vector of the signal received by the antenna subarray and the sample of the desired signal, respectively, achieving a higher signal-to-noise ratio than traditional beamforming algorithms [15].

The combination of deep learning and beamforming algorithms can better solve some tricky problems and has unique advantages in accelerating fault diagnosis efficiency and improving fault diagnosis accuracy, which not only reduces manual operation, but also makes mechanical equipment run more safely and smoothly. In order to solve the problem of the harsh working environment and serious noise interference of idlers, an MVDR-LSTM fault distance estimation model with a diagonal double rectangular array is proposed. This model can greatly suppress the interference of noise and non-coherent sound sources and realize the accurate estimation of the failure distance of idlers. At the same time, compared with the CBF-LSTM and FBF-LSTM models, our model has better robustness.

This article will first explore the relevant theory and historical background of the MVDR-LSTM roller fault distance estimation model. Secondly, the methods and processes adopted in this study will be introduced, including the specific steps of data collection and analysis. Then, we will elaborate on our research results and attempt to provide a deeper understanding of the MVDR-LSTM idler fault distance estimation model by comparing and interpreting these results. Finally, we will summarize our findings and explore their potential impact on the field of fault detection, as well as future research directions.

## 2. Theory

### 2.1. Microphone Array

A microphone array is an advanced technology used to capture sound signals that consists of multiple tiny microphones and can receive sound signals in different directions. Microphone arrays provide a more accurate and clear sound signal than a single microphone, because they suppress ambient noise and echo and can focus on receiving sound in different directions. Microphone arrays are widely used in many fields, such as video conferencing, speech recognition, and audio signal processing. In video conferencing, microphone arrays can provide clear voice signals, improving the efficiency and quality of meetings. In speech recognition, microphone arrays can improve the accuracy of speech recognition, because they reduce the impact of ambient noise. In audio signal processing, microphone arrays can realize functions such as spatial filtering and sound source localization to improve the quality of audio signals. As a result, microphone arrays have become an important part of modern communication and audio technology, and they are constantly improving our daily lives and work experiences [16].

The formation of a microphone array refers to the arrangement of microphones, and different formations can be adapted to different application scenarios. The common forms of microphone arrays are shown in Figure 1.

Line array: Multiple microphones arranged in a straight line. This array is suitable for scenarios wherein the sound source is moving horizontally.

Cross-shaped array: An array that positions multiple microphones in a cross-shaped manner, which is suitable for sound source localization and speech recognition scenarios.

Rectangular array: Multiple microphones arranged in a rectangle. This is suitable for scenarios wherein sound sources move horizontally and vertically.

U-shaped array: The microphones are arranged in a special U-shape, suitable for high-quality language recognition. For example, Translator 4.0 developed by iFlytek was paired with a U-shaped microphone array, which helped improve the accuracy and quality of sound capture. 

L-Array: Multiple microphones arranged in an L-shape. This array is suitable for scenarios wherein sound needs to be captured within a specific area.

The circular microphone array positions multiple microphones in a circle and has the following benefits: (1) 360-degree omnidirectionality. The circular microphone array can achieve 360-degree omnidirectionality in the horizontal direction and can evenly receive sound from all directions without mechanical rotation or adjusting the orientation of the microphone. This is useful for applications that need to capture the sounds from the whole environment, such as speech recognition, voice conferencing, and voice commands. (2) Eliminating noise and echo. The circular microphone array can use the delay difference and amplitude difference between multiple microphones in the array to achieve spatial sound source localization and sound source separation, so that the influence of noise and echo can be effectively eliminated. This is very helpful for collecting clear speech signals in noisy environments, thus improving the performance of speech recognition and speech processing [17,18].

The diagonal double rectangular array proposed in this article consists of two rectangles. The distance between the adjacent microphones is equal, and the centers of the two rectangles are staggered from each other, forming a diagonal line. The advantages of diagonal double rectangular arrays include: (1) The improved accuracy of sound source positioning. Since the microphones in a diagonal double rectangular array are staggered, more sound direction information can be captured, thereby improving the accuracy of sound source positioning. (2) Reduction in echo and noise interference. The diagonal double rectangular array can use the signals of multiple microphones for noise reduction processing, thereby reducing the interference of echo and noise. This is useful for speech recognition and sound source localization in some noisy environments. (3) Supporting spatial filtering. The diagonal double rectangular array can use the signals of multiple microphones for spatial filtering processing, thereby improving the quality of voice signals. (4) Supporting multi-sound source separation. The diagonal double rectangular array can realize the separation of multiple sound sources by analyzing the time domain and frequency domain characteristics of multiple microphone signals. This is of great benefit for idlers working in environments with complex sounds. In general, the diagonal double rectangular array has unique advantages in improving the positioning accuracy of sound sources, reducing echo and noise interference, supporting spatial filtering, and supporting multi-source separation. Hence, the diagonal double rectangular microphone array was selected as the acquisition array for idler fault distance signals.

### 2.2. MVDR

Beamforming technology and signal space wavenumber spectrum estimation are the two main research directions of free space signal array processing. In order to maximize the array gain, Capon proposed the MVDR beamformer, also known as the Capon beamformer, in 1969. MVDR is an adaptive beamforming algorithm based on the maximum signal-to-noise ratio (SNR) criterion. The MVDR algorithm can adaptively minimize the power in the desired direction of the array output and maximize the signal-to-drying ratio. It can greatly improve the resolution and suppress noise in spatial spectral estimation [19,20,21]. This study implemented traditional MVDR beamforming in the frequency domain based on the Acoular library. In the Acoular library, time-domain signals are first processed through the PowerSpectra class, which performs framing and windowing on the time-domain signals. Then, they are converted into frequency-domain signals through fast Fourier transform. Next, the cross spectra between signals at each frequency are calculated, and these cross spectral values are stored in a matrix. Subsequently, the SteeringVector class is used to calculate the steering vector. The steering vector contains the phase information of the microphone array relative to the signal source. This phase information is used to weight and add the signals of the microphone array to form a sound beam pointing in a specific direction. Afterwards, the cross spectral matrix and steering vector are passed to the BeamformerCapon class. Then, the signal frequency band is divided, and the envelope spectral slope value is used as an indicator to search for the optimal sub-band and select the center frequency. By setting the center frequency *f_0_* according to the frequency band division method of 1/3 octave bandwidth, the upper frequency of the 1/3 octave band $f_{h}=f_{0}\times \sqrt[{3}]{2}$, and the lower limit frequency $f_{h}=f_{0}÷\sqrt[{3}]{2}$. According to the specified center frequency and frequency band index, a synthetic beam is synthesized from the processed microphone channel data to form a sound signal, the feature matrix of the target sound signal is obtained, and the feature matrix is processed as the input layer of the LSTM model.

The specific operation steps of MVDR are as follows. Suppose the desired target location P’ in space is $\left(r_{0},\theta_{0}\right)$, the jamming signals are $i_{j}$$\left(j=1,2,3,4\cdots J\right)$, the interference location is $\left(r_{j},\theta_{j}\right)$, the array element noise is $n_{t}$, and the *nth* element signal at the receiving end is
(1)$$x\left(t\right)=a{\left(\theta_{0}\right)}_{s}\left(t\right)+\sum_{J}^{j=1}a\left(\theta_{j}\right)i_{j}\left(t\right)+n\left(t\right)$$


In this formula, the direction of the receiving guide vector a(θ) is based on $\left(r,\theta_{}\right)$, which can be expressed as
(2)$$a(\theta )=\left[1,e^{-j2\pi \frac{f_{0}d\sin(\theta )}{c}},\cdots ,e^{-j(N-1)2\pi \frac{f_{0}d\sin(\theta )}{c}}\right]$$


When the constraints are also satisfied, in order to minimize noise, the result of the objective function optimization is
(3)$$\min_{\omega }\left[\left|y{\left(t\right)}^{2}\right|\right]=\min_{\omega }\left(w^{H}Rw\right)$$

Then, the MVDR weight optimization problem can be expressed as
(4)$$\min_{\omega }\left(w^{H}Rw\right)$$
(5)$$s.t.\;\left\{\begin{array}{l} w^{H}a\left(\theta_{d}\right)=1\\ w^{H}a\left(\theta_{ij}\right)=0 \end{array}\right.$$

The essence of the MVDR beamformer is to solve the weight coefficients of each array. Then, using the Lagrange multiplier yields
(6)$$L(w)=w^{H}Rw+\lambda \left[w^{H}a\left(\theta_{d}\right)-1\right]$$

After deriving Equation (6), let it be 0, which can be obtained as
(7)$$\frac{\partial L(w)}{\partial w}=2Rw+\lambda a\left(\theta_{d}\right)=0$$

Then, it is solved as follows:(8)$$w=\mu R^{-1}a\left(\theta_{d}\right)$$

According to the MVDR criterion, the optimal value of the array weight is obtained:(9)$$w_{\mathrm{MVDR}}=\frac{R^{-1}a\left(\theta_{d}\right)}{a^{H}\left(\theta_{d}\right)R^{-1}a\left(\theta_{d}\right)}$$

### 2.3. LSTM

Recurrent neural networks (RNNs) are robust deep neural networks with high performance that process serial data of varying lengths for end-to-end distribution [22]. Long short-term memory (LSTM) is an advanced RNN architecture that includes the memory unit proposed by Hocklet and Schmiduber to solve the gradient vanishing problem in RNNs. As a time-loop network, it uses the “three-gate structure” to obtain the correlation of large-scale time series data and extract the optimal features [23,24,25]. The forgetting gate of the LSTM architecture avoids long-term dependencies and trains valuable information from historical network units to obtain more meaningful and autonomous information in sequential data. The data for the input sequence are represented as $X=[x_{1},x_{2},x_{3}\cdots x_{m}]$. Through each time step *t*, $x_{t}\in R_{s}$ is the input vector, and the hidden layer status is $h_{t}\in R^{l}$, providing the following formula:(10)$$h_{t}=f\left(h_{t-1},x_{t}\right)=f\left(Uh_{t-1}+Wx_{t-1}+b\right)$$

In this formula, $U\in R^{l*l}$, $W\in R^{l*s}$, and $b\in R^{l}$ are blended learning parameters; *f* is a nonlinear activation function; *m* is the sequence length; s is the size of the input; and *l* is the hidden size.

A diagram of the LSTM unit is shown in Figure 2a with hidden state variables $h_{t}$. LSTM retains the encoding time phase *t* of the storage unit *c_t_*. The efficiency of the storage unit is determined by three gates: the input gate *i_t_*, output gate *o_t_*, and forgetting gate *f_t_*. The upgraded equation can be expressed as:(11)$$i_{t}=sigm\left(U_{i}h_{t-1}+W_{i}x_{t}+b_{i}\right)$$
(12)$$f_{t}-sigm\left(U_{f}h_{t-1}+W_{f}x_{t}+b_{f}\right)$$
(13)$$o_{t}=sigm\left(U_{o}h_{t-1}+W_{o}x_{t}+b_{o}\right)$$
(14)$${\tilde{c}}_{t}=\tanh\left(U_{c}h_{t-1}+W_{c}x_{t}+b_{c}\right)$$
(15)$$c_{t}=f_{t}⊙c_{t-1}+i_{t}⊙{\tilde{c}}_{t}$$
(16)$$h_{t}=o_{t}⊙\tanh\left(c_{t}\right)$$
where $U\in R^{l*l}$, $W\in R^{l*s}$, and $b\in R^{l}$ are the learning optimization parameters; *sigm* is an S-type function; tanh is the hyperbolic cut function; and operator ⊙ represents the element-level product.

The main operating points of the LSTM unit are defined according to each phase of the time *t* and comprise the following three steps: (1) From the new input functions *x_t_* and early hidden state *h_t−1_*, one can obtain the forgetting gate *f_t_*, whose values range from 0 to 1. When the value of *f_t_* reaches 1, the last memory cell *c_t−1_* of incoming data is strongly preserved. On the other hand, when the value of *f_t_* reaches 0, the incoming data are distributed and cannot be retained. (2) Input gates can be removed from the new input functions *x_t_* and early hidden state *h_t_* and added to the memory cell, specifying *c_t_*. (3) Therefore, the output gate must select an item from the memory cell that can be processed to form a new hidden state *h_t_* [26,27,28,29].

The overall LSTM model is shown in Figure 2b. The LSTM model has two hidden layers, an input layer, and an output layer. When viewed separately, it is a simple BP neural network, but as time passes, the hidden layer information *h*, *c* also begins to be processed, forming a complete LSTM model.

## 3. Simulation

### 3.1. MVDR Algorithm Simulation

The MVDR algorithm was simulated by MATLAB, and the parameters set are shown in Table 1. The desired signal angle was set to 10, and two interfering signals were added at angles of −30 and 30. The simulation analysis showed that the waveform signal with an angle of 10 was convex, and the angle in the −30 and 30 directions was effectively suppressed. It can be seen that the effect of MVDR in the direction of noise suppression was obvious. See Figure 3.

### 3.2. Microphone Array Formation Selection

The simulation results of the microphone array are shown in Figure 4, and the sound source position of the diagonal double rectangular array was closest to the true value and the suppression effect on the side lobes. The main lobe of the uniform straight array was narrower but it covered a greater area, and the results were less accurate. Although the position of the main sound source could be accurately located, the interference from other sound sources appeared, and the suppression effect on the side lobe was not ideal, producing an error. The cross-shaped array was relatively uniform, and the main lobe of the linear array was more prominent, but there was still a problem of inaccurate positioning. Due to the existence of too many main lobes, the rectangular array had the most obvious inhibition effect on the side lobes, but the localized main lobe area was too large to play the role of precise positioning. Therefore, the proposed double rectangular diagonal array had the best robustness and was most suitable as a microphone array for an idler failure distance estimation model.

## 4. Experiments

### 4.1. LSTM Model Parameter Settings

The LSTM model consisted of one input layer, two hidden layers, one fully connected layer, and one output layer. When the hidden layer was set to one layer, the effect achieved was not ideal. After increasing it to two layers, the demand was already met. When increasing it to three layers, the calculation time of the model increased. Therefore, two hidden layers were used. The accuracy of the model with different numbers of hidden layers is shown in Figure 5. We used the cross-entropy loss function as the objective function to guide the learning of network parameters and selected Softmax as the activation function. Softmax is an activation function specifically designed for multi-class problems. It can convert a set of real numbers into an output and a probability distribution of 1, which is very suitable for the output layer probability prediction of multiple-classification problems. As the optimization method, we selected Adam. Adam optimizers are some of the most popular optimizers in deep learning. They are suitable for a wide variety of problems, including models with sparse or noisy gradients. Their ability to be easily fine-tuned makes it possible to obtain good results quickly; in fact, the default parameter configuration usually achieves good results. Adam optimizers combine the best of AdaGrad and RMSProp. Adam uses the same learning rate for each parameter and adapts independently as the learning progresses.

### 4.2. Experimental Setup

The diagonal double rectangular microphone array was used as a sound signal acquisition device. The sampling frequency was 16,000 Hz, the number of sampling points was 1024, and the device consisted of 8 MEMS microphones and microphone holders. In the experiment, five positions were selected as sound source points. The distance between each sound source points was 0.5 m, and the microphone array was 1 m away from the ground and 1 m away from the central sound source, as shown in the schematic diagram in Figure 6.

### 4.3. Experimental Process

The flow chart of the experiment is shown in Figure 7, and the specific experimental flow is as follows:Step 1—The microphone array was used to collect the idler fault audio signal.Step 2—MVDR processing was performed on the collected signal to generate idler fault distance data.Step 3—We processed the data again to build the dataset.Step 4—We divided the data into the training set (90%) and testing set (10%).Step 5—We trained and tested the model to draw conclusions.

### 4.4. Background Noise Interference Experiment

White noise was used as background noise to collect the sound signal of the idler, and the waveform diagram obtained is shown in Figure 8. The fault characteristic information of the idler was masked by the white noise, so as to imitate the environment during on-site inspection.

After beamforming the acquired sound signal, the processed data were passed to the LSTM model. The data samples were processed 100 times, and the number of data (samples) passed to the program for training at a time (batch_size) was 16. The accuracy and loss rates of the training set and the test set were observed as the model performance, and the accuracy of the training and test sets of the MVDR-LSTM model gradually converged after 65 iterations, with the accuracy remaining at 100% for both. The loss rate remained steady after 65 iterations as the number of iterations increased, and the results are shown in Figure 9.

### 4.5. Impact Noise Interference Experiment

Under the same experimental conditions, the background noise was replaced with irregular impact interference noise. The obtained waveform is shown in Figure 10, and the impact noise covered part of the fault characteristics, showing a waveform with higher amplitude, which increased the difficulty of estimating the fault distance of the idler.

Similarly, 100 iterations were performed on the newly generated data samples, and the number of data (samples) passed to the program for training (batch_size) was 16. The accuracy and loss rates of the training and test sets are shown in Figure 11. The accuracy of the training and test sets of the MVDR-LSTM model converged after 80 iterations, and the loss rate converged after 85 iterations.

### 4.6. Evaluation Criteria and Comparison of Models

#### 4.6.1. Data Comparison for Diagonal Double Rectangular Microphone Array

In order to verify the superiority of the proposed model, the diagonal double rectangular array was used as the acquisition device, and the MVDR-LSTM, CBF-LSTM, and FBF-LSTM models were used as the comparison models. We set the number of data (samples) passed to the program for training at a time (batch_size) as 16, the activation function was Softmax, and the optimizer was Adam. Under background noise interference, the accuracy of the three models, the confusion matrix classification results, and KPCA dimensionality reduction analysis were used as evaluation criteria. The accuracy results obtained over 10 experiments are shown in Figure 12. The accuracy of the MVDR-LSTM model was higher and more stable than that of the CBF-LSTM and FBF-LSTM models, with average accuracy values of 99.74%, 94.60%, and 74.69%, respectively. The highest values were 100%, 100%, and 86.78%, and the lowest values were 98.32%, 89.52%, and 60.33%, respectively. Based on the data of the 10 experiments, we found that the average accuracy of the MVDR-LSTM model was higher than that of the other two models.

We used a confusion matrix to compare the classification results with the actual measured values and displayed the accuracy of the classification results in the same confusion matrix. As shown in Figure 13, we found that the MVDR-LSTM model achieved perfect prediction results in all categories without misclassification. The CBF-LSTM model had some classification errors, while FBF-LSTM demonstrated more misclassification, so the MVDR-LSTM model was more effective.

The KPCA dimensionality reduction diagram of the idler fault distance data sample is shown in Figure 14. The feature expression distribution learned by the MVDR-LSTM model showed the clearest boundary, so the extracted features were more easily segmented, which meant that it was easier to classify each fault distance category. There was much evidence of feature misclassification by the CBF-LSTM and FBF-LSTM models, and the effect was not good.

#### 4.6.2. Data Comparison for Circular Microphone Array

The MVDR-LSTM, CBF-LSTM, and FBF-LSTM models in the circular array were selected as comparison models. We set the number of data (samples) passed to the program for training at a time (batch_size) as 16, the activation function was Softmax, and the optimizer was Adam. Under background noise interference, the accuracy of the three models, the confusion matrix classification results, and KPCA dimensionality reduction analysis were used as evaluation criteria. The accuracy results obtained over 10 experiments are shown in Figure 15. The accuracy of the MVDR-LSTM model was higher and more stable than that of the CBF-LSTM and FBF-LSTM models, with average accuracy values of 89.72%, 85.33%, and 74.53%, respectively. The highest values were 93.33%, 93.33%, and 82.32%, and the lowest values were 86.78%, 54.28%, and 60.00%, respectively. Based on the data of 10 experiments, we found that the average accuracy of the MVDR-LSTM model was higher than that of the other two models.

The confusion matrix of the roller failure distance data sample of the circular array is shown in Figure 16. We found that the sample classification results of the MVDR-LSTM model included less misclassification, and there were further advantages over the other two models.

The KPCA dimensionality reduction diagram of the idler fault distance data sample obtained by the circular array acquisition signal is shown in Figure 17. Although the boundary shown by the feature distribution expression of the three types of models was not obvious, the classification effect of MVDR-LSTM was the best, and it demonstrated less data misclassified compared with the other two models.

Through the comparative analysis of the above methods, we observed that the effect of acquiring signals using a diagonal double rectangular array in a noisy interference environment was better than when using a circular array, and the MVDR-LSTM model presented substantial advantages in all aspects compared with the other five models, so the MVDR-LSTM model was selected as the fault distance estimation model.

## 5. Conclusions

In this paper, an MVDR-LSTM model for idler fault distance estimation was proposed. The model was tested in two different interference environments, and the results were compared with five other models to reach the following conclusions:Due to the special structure of an idler, the human diagnosis steps are cumbersome and time is wasted. Based on deep learning, our model removed the tedious step of manually extracting idler fault features, which could greatly shorten the time required and improve the efficiency of idler fault diagnosis.Through simulation experiments, we found that the diagonal dual rectangular microphone array had higher resolution and noise resistance compared to the other microphone arrays and could be beneficial to the estimation of roller fault distance.After adjusting the model structure and parameters, we trained the generated idler fault distance samples. The accuracy of the proposed model was 100%, with better performance than the other models and results closer to the true values.Five fault locations during idler operation were analyzed, and the experimental results showed that the proposed model had the ability to estimate the fault distance and provide better robustness. It could be used as a standard for judging the fault distance of idlers and provide ideas for combining beamforming algorithms and deep learning with idler fault diagnosis.

The main advantage of LSTM is its ability to learn and remember long-term sequence dependencies. However, compared to CNNs, LSTM models have higher computational complexity because they contain gating mechanisms and time-step dependencies, resulting in greater computational costs in the training and inference process, which may require a longer training time and higher computing resources. Thus, it is necessary to continue to optimize the model in future research to reduce the computational costs and shorten the computing time.

## Figures and Tables

**Figure 1 sensors-23-05094-f001:**
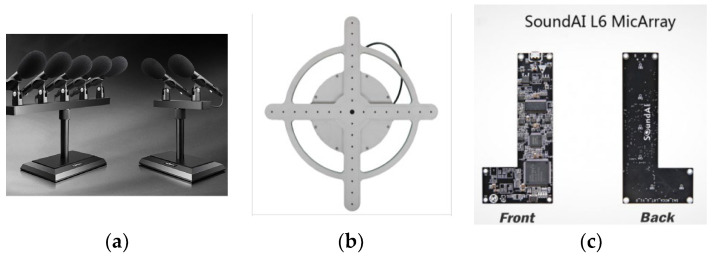
Microphone array formations: (**a**) line array, (**b**) cross-shaped array, (**c**) L-array.

**Figure 2 sensors-23-05094-f002:**
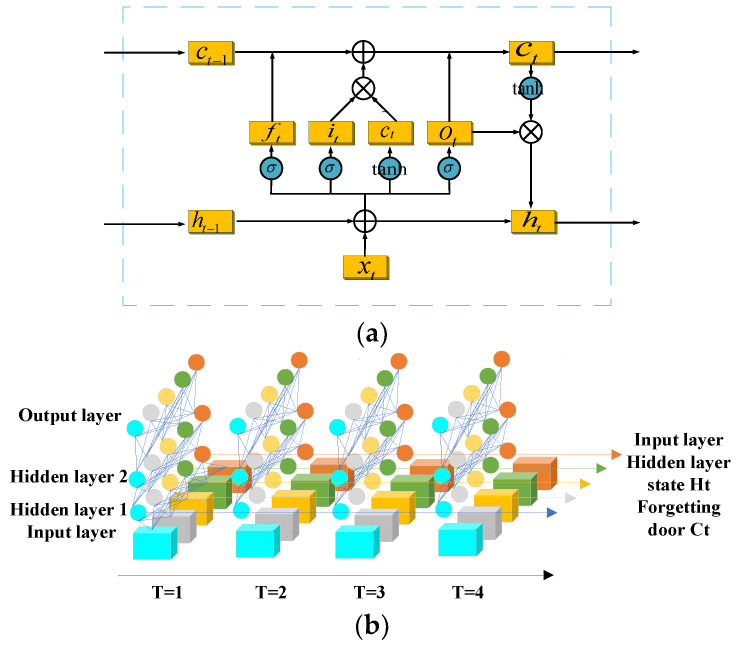
LSTM model: (**a**) LSTM cell model, (**b**) schematic diagram of the overall LSTM model.

**Figure 3 sensors-23-05094-f003:**
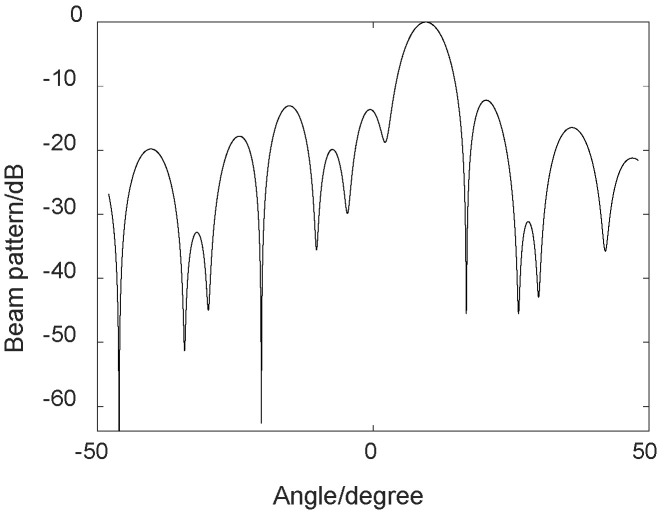
MVDR simulation.

**Figure 4 sensors-23-05094-f004:**
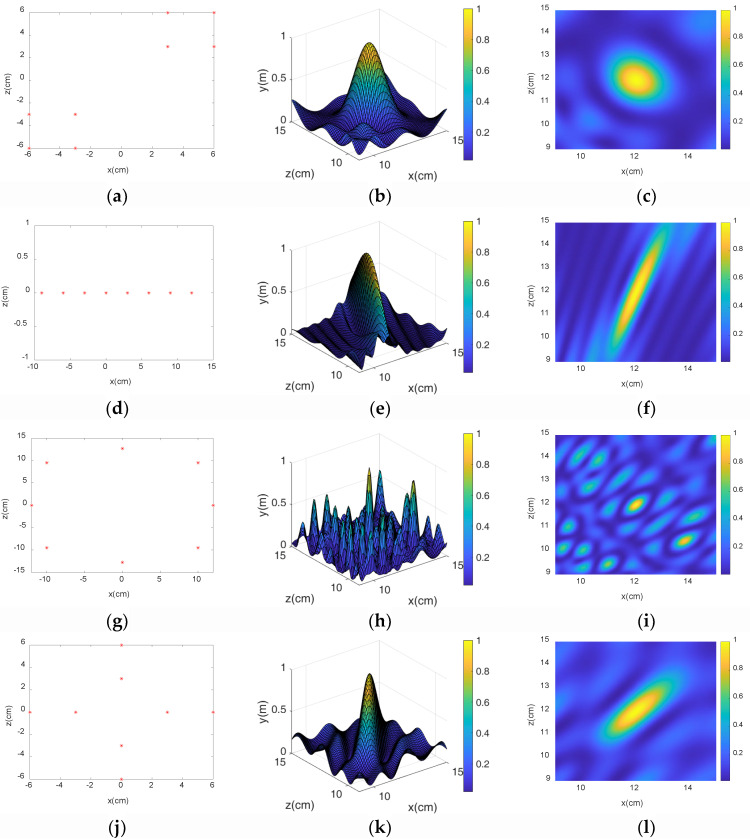
Simulation results of each array: (**a**–**c**) schematic diagram of diagonal double rectangular array and two- and three-dimensional positioning diagrams, (**d**–**f**) linear array diagram and two-and three-dimensional positioning diagrams, (**g**–**i**) circular array schematic diagram and two- and three-dimensional positioning diagrams, (**j**–**l**) cross array schematic diagram and two- and three-dimensional positioning diagrams, (**m**–**o**) rectangular array schematic diagram and two- and three-dimensional positioning diagrams.

**Figure 5 sensors-23-05094-f005:**
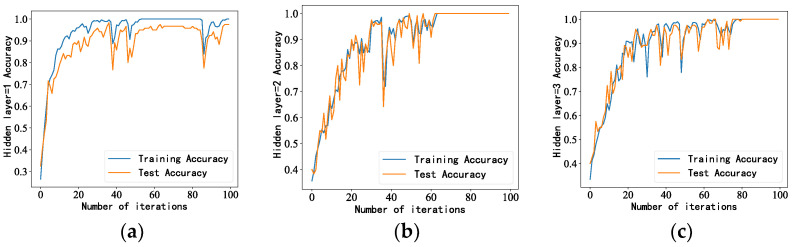
Model accuracy under different numbers of hidden layers: (**a**) Accuracy when the number of hidden layers is 1, (**b**) Accuracy when the number of hidden layers is 2, (**c**) Accuracy when the number of hidden layers is 3.

**Figure 6 sensors-23-05094-f006:**
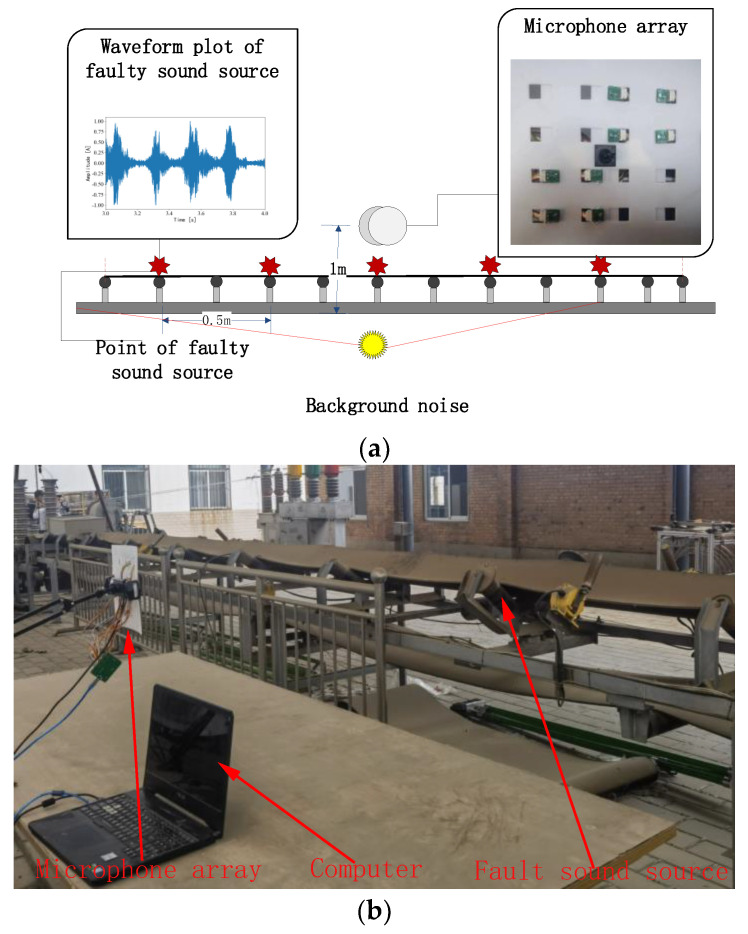
Schematic diagram of the experimental layout: (**a**) experimental schematic diagram, (**b**) field experiment diagram.

**Figure 7 sensors-23-05094-f007:**
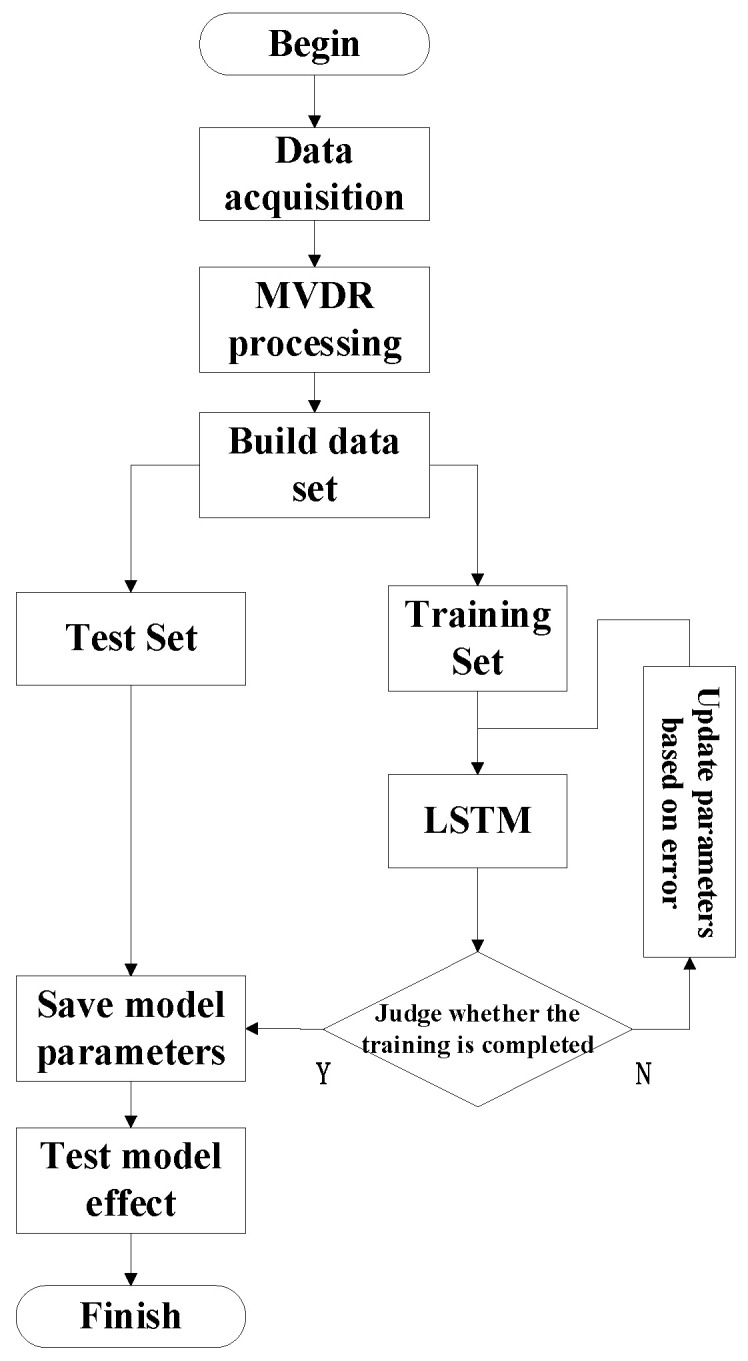
Experimental flowchart.

**Figure 8 sensors-23-05094-f008:**
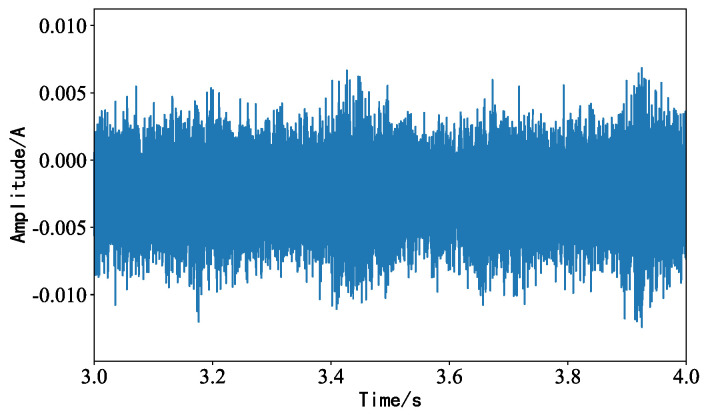
Roller acquisition signal waveform under white noise interference.

**Figure 9 sensors-23-05094-f009:**
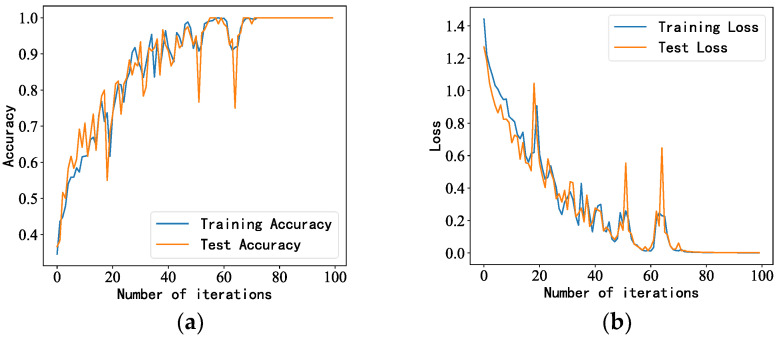
LSTM model performance under background noise interference: (**a**,**b**) accuracy and loss rate of training and test sets, respectively.

**Figure 10 sensors-23-05094-f010:**
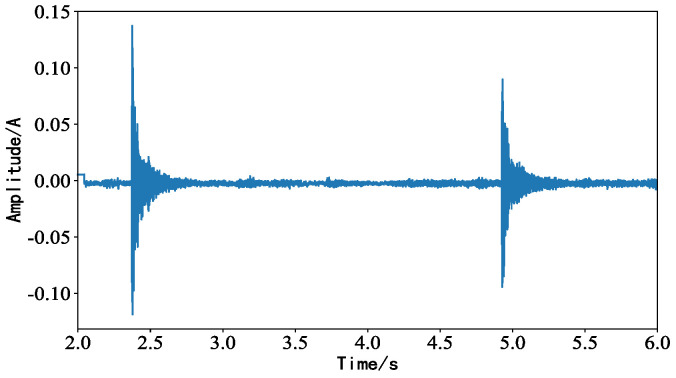
Waveform diagram of idler acquisition signal under impact noise interference.

**Figure 11 sensors-23-05094-f011:**
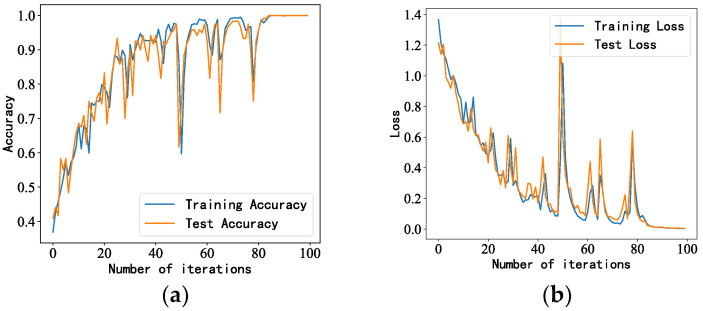
LSTM model performance under impact noise interference: (**a**,**b**) accuracy and loss rate of training and test sets, respectively.

**Figure 12 sensors-23-05094-f012:**
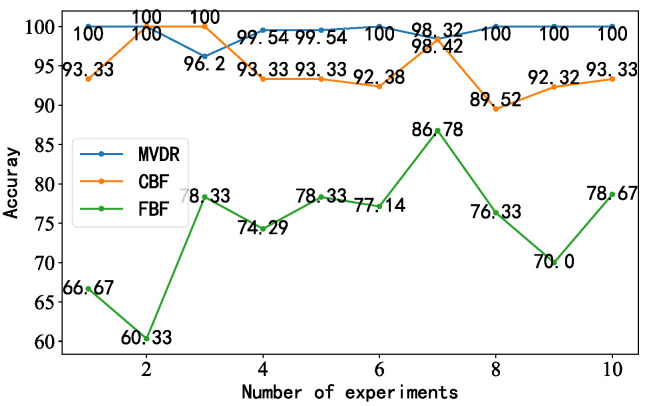
Line chart illustrating the accuracy of the three models for the diagonal double rectangular array.

**Figure 13 sensors-23-05094-f013:**
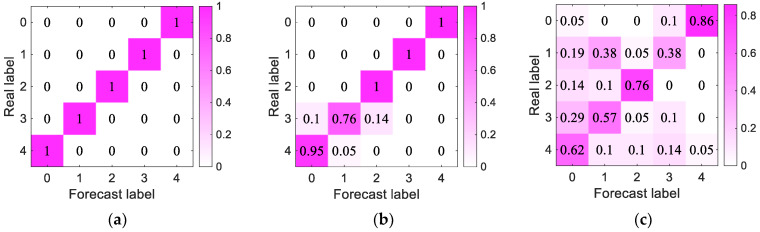
Diagonal double rectangular array per-model confusion matrix results: (**a**–**c**) MVDR-LSTM, CBF-MVDR, and FBF-LSTM confusion matrixes, respectively.

**Figure 14 sensors-23-05094-f014:**
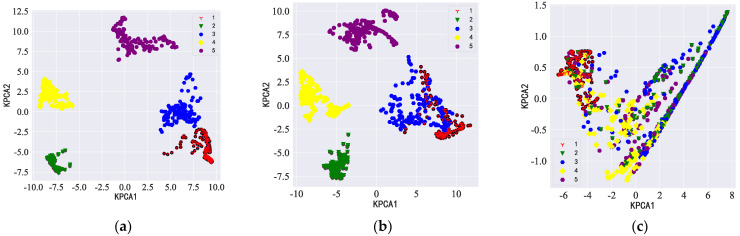
KPCA dimensionality reduction diagram for each model with the diagonal double rectangular array: (**a**–**c**) MVDR-LSTM, CBF-MVDR, and FBF-LSTM KPCA dimensionality reduction analyses, respectively.

**Figure 15 sensors-23-05094-f015:**
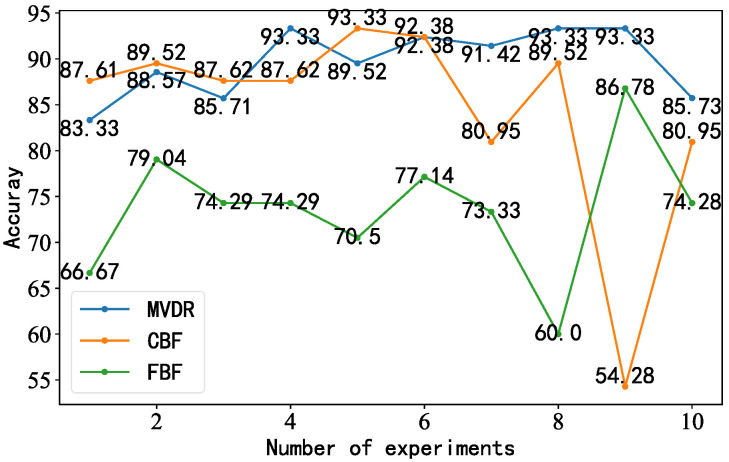
Line chart illustrating the accuracy of the three models for the circular array.

**Figure 16 sensors-23-05094-f016:**
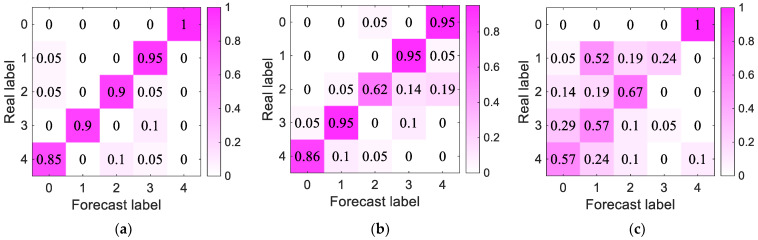
Circular array per-model confusion matrix results: (**a**–**c**) MVDR-LSTM, CBF-MVDR, and FBF-LSTM confusion matrixes, respectively.

**Figure 17 sensors-23-05094-f017:**
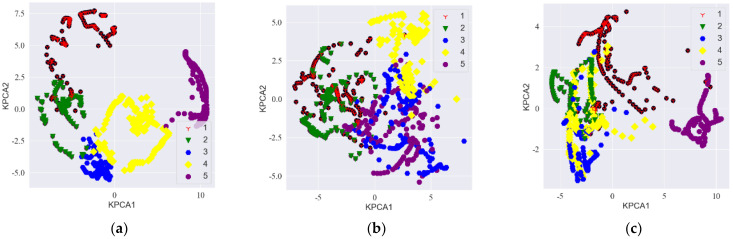
KPCA dimensionality reduction diagram for each model with the circular array: (**a**–**c**) MVDR–LSTM, CBFMVDR, and FBF-LSTM KPCA dimensionality reduction analyses, respectively.

**Table 1 sensors-23-05094-t001:** Simulation parameters.

Parameter Name	Parameter Value
Number of arrays	18
Desired signal angle	10
Interference signal angle	−30, 30
SNR	10
INR	10
Number of stories	100

## Data Availability

Not applicable.

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
