# Peer review of "MVDR-LSTM Distance Estimation Model Based on Diagonal Double Rectangular Array"

_sensors, 2023, doi:10.3390/s23115094_

Round 1
Reviewer 1 Report
I have examined manuscript entitled “MVDR-LSTM distance estimation model based on diagonal double rectangular array”, and made comments as follows:
From the whole, this paper proposes a MVDR-LSTM distance estimation model to realize roller fault diagnosis of belt conveyor. The technical route is clear. However, the advantage of minimum variance distortionless response (MVDR) should be stated clearly in the Introduction part.
Reviewer 2 Report
Dear Authors,
In the beginning I want to rise two points:
Paper is devoted to important and high impact area of predictive maintenance.
Secondly usage of AI techniques in area of engineering is remarkably interesting and provide high outcome results.
Finally, paper is well structured, properly written, but some grammar, style, syntax and typo mistakes are encountered. Examples:
1. Abstract: "robustness than the conventional beamforming" - underlined text is written with incoherent font.
2. Equation (3) and (4). Besides it is well known function, I will suggest that min function take arguments into parenthesis such as min( arguments ). This syntax gaining popularity based on similarity to almost all programing language syntax where functions have parenthesis for arguments.
3. Paragraph 4.1 "4.1LSTM model" - no space.
4. I was taught that after value and before unit single space is given. Paragraph 4.2, examples: "0.5m", "1m". I would like to raise this point for your consideration.
5. Paragraph 4.3: There is no plain text inside. No introduction, steps are part of paragraph or Figure 6.?
6. Paragraph 4.4: Even subparagraph should not be started from Figure.
7. Paragraph 4.6, incl. 4.61, 4.62 - incoherent numbering and font for subchapter headings.
8. Figure 12, 13, 14, 15 are very blurry. It is hard to read values and reader must pay a lot of attention to realize what they show.
Remarks, which I mentioned above are just only examples of mistakes and you should proceed carefully proofread, nonetheless they are simple to correct and even some of them are only for your consideration.
At this point I would like to share with you some of my serious concerns/remarks:
1. Main assumption of publishing the papers, especially in science and engineering is a providing whole knowledge for reader, which may try to reproduce your setup and obtain same results. Here this is missing. Based on your paper so many white spots are missing, and you should provide more details about your workflow. You can always add i.e., link to github repository.
2. Many arbitrary assumptions without comment. I know that in engineering world it is common use arbitrary assumptions, coefficients, approaches etc. But it is good when it is described in straight way.
3. When you mention about acquisition of signal from real object, in short, you are performing experiment on existing objects, I always require photo of the object and measurement setup. Same for Chapter 2.1 - you describe microphone array while there is no photo of the microphone array.
4. Conclusions are correct however you made huge shortcut from your results to general (correct anyway) statements. Please rewrite it in way where you describe reasoning from your direct results to your general conclusions.
Level of English language overall is high, however carefully proofread should be done especially for longer sentences.
Reviewer 3 Report
This paper proposes a MVDR-LSTM model for idler fault distance estimation using a diagonal double rectangular microphone array. The model is compared with five other models and found to have better robustness and accuracy. The use of deep learning algorithms and beamforming techniques improves fault diagnosis efficiency and reduces manual operation, making mechanical equipment run more safely and smoothly. The model greatly suppresses noise interference and non-coherent sound sources, enabling accurate estimation of the failure distance of idlers. Overall, this study demonstrates the potential of combining deep learning and beamforming algorithms for solving tricky problems in fault diagnosis.However, there are some problems, such as references and experimental verification, etc. Detailed comments are listed as follows. Hope it can provide help for improving the quality of this paper.
1) The references to this article are insufficient. The following reference is recommended to be added.
(1) Jian Yu, Yu Wen, Lei Yang*, Zhibin Zhao**, Yanjie Guo, Xiao Guo, Monitoring on Triboelectric Nanogenerator and Deep Learning Method, Nano Energy, 2022, 92, 106698.
2、 How does the MVDR-LSTM model compare to other models in terms of computational efficiency and memory requirements?
3、Can the proposed model be applied to other types of machinery beyond idlers, and if so, what modifications would need to be made?
4、How does the diagonal double rectangular microphone array compare to other microphone arrays in terms of cost and ease of implementation?
Reviewer 4 Report
In this manuscript, a bearing fault sound source distance estimation method is proposed. The angular double rectangular microphone array is used as the acquisition device, and the minimum variance distortionless response (MVDR) and long short-term memory network (LSTM) are used as the processing models. The effect of the proposed method is verified through experimental signals. The method proposed in the manuscript has some novelty, but there are still the following issues that need to be revised.
1. It seems that there is a missing parameter in the LSTM cell model in Figure 1.
2. In the parameter settings of the LSTM model, the hidden layer is set to 2 layers, what is the basis for the setting.
3. The activation function in the LSTM unit diagram is tanh, and softmax is selected as the activation function. What is the basis for selection?
4. The objective function of the model is not explicitly stated in the manuscript. Please add description.
5. The numerical value of batch_size is selected as 16, what is the basis.
6. In the conclusion, it is mentioned that CNNs are superior to LSTMs in terms of computational complexity, why not use CNN as a deep learning model, but instead use LSTM? The advantages of LSTM compared to CNN should be discussed in the manuscript as necessary.
7. In the experimental part, the description of experimental equipment, experimental environment, signal acquisition process, sampling frequency, etc. is not specific enough.
8. It is recommended to carefully proofread the layout, formulas, and grammar in the manuscript. The parameters in the formula should be explained, and the formula number and figure number should be mentioned in the text.
9. The figures description in the article needs to be carefully checked. The experimental schematic shown in Figure 5 was mistakenly described in the manuscript as Figure 7, while Figure 7 lacks a description.
10. In the introduction of MVDR, the letters used to represent the center frequency should be consistent with the letters in other formulas.
11. Recently, there are new trends of rolling bearing fault diagnosis via energy harvesting techniques. It is suggested to add description in Introduction on relative works such as: Energy, 2022, 238, 121770; Energy Conversion and Management, 2019, 180, 811-821.
12. The conclusion section suggests further optimization to highlight the advantages and limitations of the methods proposed in the manuscript.
13. The outline of the manuscript is suggested to be supplemented in the last paragraph of the introduction.
The overall quality of English is fine. However, there are some grammar and typo errors. Please check the manuscript carefully and revise them.
Round 2
Reviewer 4 Report
The manuscript is well revised and can be accepted for publication now. Thanks.